# Evaluation of Aquaculture Water Quality Based on Improved Fuzzy Comprehensive Evaluation Method

Guodong You [1,*,†] , Bin Xu [1,†], Honglin Su [1,†], Shang Zhang [1,†], Jinming Pan [2,‡], Xiaoxin Hou [1], Jisheng Li [1] and Runsuo Ding [3]

1   College of Electronic Information and Automation, Tianjin University of Science and Technology, Tianjin 300222, China; binxu@mail.tust.edu.cn (B.X.); suhonglin@mail.tust.edu.cn (H.S.); sunshineyang603@sina.com (S.Z.); houxiaoxin@tust.edu.cn (X.H.); lijish@tust.edu.cn (J.L.)
2   College of Biosystems Engineering and Food Science, Zhejiang University, 866 Yuhangtang Road, Hangzhou 310058, China; yougdong@163.com
3   Tianjin Institute of Agricultural Machinery, Tianjin 300193, China; yougdong@sina.com
*   Correspondence: yougdong@tust.edu.cn; Tel.: +86-131-3229-9319
†   These authors contributed to the work equally and should be regarded as co-first authors.
‡   The author is second author.

**Abstract:** The quality of aquaculture waters is directly related to water management and aquaculture efficiency, which puts forward higher requirements for water quality evaluation. Based on the consideration of the influence of temporal and spatial changes on the water quality, this paper proposed an improved fuzzy comprehensive evaluation method for aquaculture water quality evaluation. Specifically, constructing a new membership function in the first place, and then selecting dissolved oxygen, pH, temperature and ammonia nitrogen content as water quality indexes for aquaculture, after that, collecting 60 sets of water quality index for different seasons in the past three years, finally, evaluating the water quality of Yangjiabo Aquaculture Base. Meanwhile, comparing it with the evaluation results of the single factor evaluation method and the traditional fuzzy evaluation method. The results show that the water quality of the Yangjiabo Aquaculture Base is at the worst level in winter, and the water quality has improved significantly in spring, summer and autumn. Compared with the other two method, the improved method can comprehensively reflect the changes in water quality with time and space, which is more practical, and so it can be considered to provide a scientific basis for efficient aquaculture and water quality classification.

**Keywords:** time dimension; weight reconstruction; evaluation factor; model test





## 1. Introduction

For aquaculture, the quality of aquaculture water directly affects the quality of aquatic products [1–3]. In order to ensure the sustainable use of water quality in aquaculture, scientific management must be carried out, and water quality assessment is an important means of scientific management. Water quality evaluation needs to select water quality indexes and corresponding evaluation standards, then determine the importance of each index through scientific calculation method, and finally evaluation the comprehensive quality grade of the water.

Aquaculture water quality index mainly includes dissolved oxygen (DO), pH value, temperature, ammonia nitrogen content [4–6]. Zhang Xianyu et al., researched the water quality of pond and cage-aquaculture areas in Zhuanghe River, China, and the results show that among multiple indicators, dissolved oxygen, water temperature and pH value are the most important factors affecting the water quality [7]. Akyol O et al., monitored species richness of the fish farms in the Aegean Sea [8]. The results showed that water temperature, pH value and dissolved oxygen content were the three main factors that determined fish richness. M.T. Jafari et al., used corona discharge ion mobility spectrometry (CD- IMS) [9],

and Shi Jiahui used microfluidic technology and colorimetric spectroscopy to detect the content of ammonia nitrogen quickly and accurately [10], and optimize the evaluation of water quality. Wu Kai et al. using one-way analysis of variance (one-way ANOVA) and Least-Significant Difference (LSD), analyzed the changes of water quality in different water layers in the ecological crab breeding pond [11], and proved that the concentration of dissolved oxygen in water was negatively correlated with the concentration of $NH_4^+ - N$ and $NO_3^- - N$. In order to accurately understand the water quality and major pollution factors of the direct inflow of Danjiangkou Reservoir in China, Xia Fan et al. using the Single Factor Assessment Method, Comprehensive Pollution Index Method, and Principal Component Analysis (PCA) to evaluate water quality [12], the evaluation show that the Single Factor Assessment Method only gives evaluation category, the Comprehensive Pollution Index Method and PCA is suitable for different space-time change of water quality. Sung Eun Kim et al. evaluated the changes of water quality in 28 monitoring stations of Nakdong River in Korea by combining Exploratory Factor Analysis (EFA) with PCA method of Empirical Orthogonal Function (EOF) [13], this method plays a positive role in the spatial and seasonal evaluation of Nakdong River monitoring network. In addition, Zhenya Li et al. proposed an improved Informative Weighting and Ranking (TIWR) Method based on Technique for Order Preference by Similarity to Ideal Solution (TOPSIS) [14], which has certain guiding significance for water environment protection and management. Gonzalo Carrasco et al. used multivariate statistical HJ-Biplot method [15] and Franciska M. Schets et al. used Integrated Water Quality Index (IWQI) model [16] to test and evaluate the water quality of fish culture circulation system and swimming pool respectively. The results show that this method is practical for the scientific and reasonable evaluation of water quality.

In view of the fuzziness and uncertainty in the evaluation of water quality problems, Lei Liping et al. combined the entropy weight method with the cloud model to construct the entropy weight-normal cloud model [17], and used the entropy weight method to determine the weight, avoiding the influence of subjective factors in the evaluation process, as a qualitative and quantitative conversion tool, the cloud model can comprehensively consider the fuzziness and randomness in the evaluation process. The results of surface water quality evaluation are analyzed and compared with the results of single-factor evaluation and fuzzy comprehensive evaluation, the rationality and scientific nature of the model are proved. Yilmaz Icaga proposed an evaluation index model of surface water quality classification based on fuzzy logic [18], summing up the quality parameters of different concentrations with fuzzy rules, and de-fuzzy the summation values. The water quality of Eber Lake in Turkey was evaluated, which proved the practical application and feasibility of this method. Xiaojing Wang et al. proposed a fuzzy similarity measurement method [19] to judge the closeness between two fuzzy sets and evaluate the water quality of the Haihe River in China. Mariangela Dutra de Oliveira et al. connected the fuzzy logic with the new RWQI method [20], carried out experimental verification in 24 water source data sets in Brazil, and evaluated its correlation with Raw Water Quality Index (RWQI), Water Quality Index (WQI_CETESB), treated water turbidity and coagulant dose, the results proved the rationality of the Raw Water Quality Index Fuzzy (RWQIF) tool. In order to comprehensively evaluate the Chinese Guangzhou flow river irrigation area water environment quality, Xian Qun Jiang et al. used fuzzy neural network based on T-S model, the authors made an evaluation of the water quality characteristics of the middle route of south-to-north water transfer and analysis [21], the dissolved oxygen, chemical oxygen demand, ammonia nitrogen six indicators, has an important influence on the water quality of permanganate index, total phosphorus and total nitrogen in the screening, the results showed that the model evaluation level of water quality change trend and the change trend of index data. In addition, Xiaojing Wang et al. also constructed a fuzzy water quality evaluation model to evaluate the water quality of Jialing River in China. Jyotiprakash G. Nayak et al. assessed the water quality of Godavari River in India by using fuzzy reasoning methods [22], which ensured the stability of aquaculture water quality.

Zhang Qian et al. collection of Erhai Lake in China in 1922–2015 water quality monitoring data, using the hierarchical analysis and entropy weight method combined with the fuzzy comprehensive evaluation method [23], using multiple water quality index, starting from the two aspects of water quality and eutrophication of lakes, evaluate the Erhai Lake water environment change trend for China, and compared with single factor evaluation results. The results show that compared with the single factor evaluation, the improved fuzzy comprehensive evaluation method can better reflect the dynamic characteristics of water quality in Erhai Lake in the last 20 years. Hou Yuting et al. in Wujiang River Basin in China, such as karst mountainous area water quality evaluation as an example, the chroma of water pollutants index algorithm is applied to the double excess weight method, and the standard method of multiple combination new comprehensive weighting, to build an improved model of fuzzy comprehensive evaluation method with double weight exceeding standard weighting method, exceeding the standard multiple method and the method of grey correlation number instead of fuzzy membership comprehensive evaluation method to compare analysis [24], provide a new research method for water quality evaluation. Gao Xueping et al. proposed the concept of temporal distribution weight matrix and combined the measured weight vector with temporal distribution matrix to give combined weight and obtain the comprehensive weight vector [25]. The results obtained when this method is applied to a diversion channel water quality assessment are more in line with the actual measurements as compared to the traditional method.

Due to the impact of the surrounding environment and climate, the water quality of aquaculture waters often changes regularly, and the importance of water quality indexes also changes accordingly. In terms of time, this change is not only manifested as seasonal cycle change but also correlated with water quality indexes on the same day in different years, that is, different seasons and different years correspond to different index weights on the same day. The impact on water quality is mainly reflected in monsoon, temperature and rainfall (non-point source pollution), the wind direction and wind speed in the aquaculture water area will also change accordingly, thus affecting water quality. In view of this, this paper proposed a method of aquaculture water quality evaluation based on improved fuzzy comprehensive evaluation method. Compared with the traditional fuzzy comprehensive evaluation method, this method improves the evaluation standard set, reconstructs the weight of water quality evaluation factor in the time dimension, and finally obtains the comprehensive weight vector. Through experiments, the evaluation results were compared with the single factor evaluation method and the traditional fuzzy evaluation method, which verified the effectiveness and scientificity of the method for the evaluation of the water quality in aquaculture waters.

## 2. Methods and Materials

### 2.1. Traditional Fuzzy Comprehensive Water Quality Assessment Method

The basic idea of the traditional fuzzy comprehensive water quality evaluation method is as follows. Firstly, to form a fuzzy matrix, it is needed to establish the monitoring data of each water quality factor index and the membership standards of each level. Then, multiplying the factor weight vector by the fuzzy matrix to obtain a comprehensive water evaluation data set [26]. The traditional fuzzy comprehensive water quality evaluation method [27,28] is divided into five steps:

Step 1: Build the evaluation factor subset. The evaluation factors selected are also different for different evaluation objects. In the evaluation of aquaculture water quality, evaluation factors are mainly selected: dissolved oxygen, pH value, temperature, ammonia nitrogen content, water level, turbidity.

Step 2: Determine evaluation criteria. Different from the first step, the evaluation criteria are different for different evaluation objects. In the evaluation of aquaculture water quality, the evaluation standard is Fishery Water Quality Standard (GB11607-89) [29].

Step 3: Determine membership function and construct fuzzy matrix.

According to the effect mechanism of evaluation factors on the water quality of aquaculture [30], the membership function of evaluation factors can be divided into two types, namely increasing type and decreasing type.

(1) Incremental type

When $x_i \leq S_{i,j}$, the expression is:

$$\begin{cases} r_{i,1} = 1 \\ r_{i,2} = r_{i,3} = r_{i,4} = r_{i,5} = 0 \end{cases} \tag{1}$$

When $S_{i,j} \leq x_i \leq S_{i,j+1}$, the expression is:

$$\begin{cases} r_{i,j} = \frac{S_{i,j+1} - x_i}{S_{i,j+1} - S_{i,j}} \\ r_{i,j+1} = 1 - r_{i,j} \end{cases} \tag{2}$$

When $S_{i,5} \leq x_i$, the expression is:

$$\begin{cases} r_{i,5} = 1 \\ r_{i,1} = r_{i,2} = r_{i,3} = r_{i,4} = 0 \end{cases} \tag{3}$$

(2) Decremental type

When $x_i \geq S_{i,1}$, the expression is:

$$\begin{cases} r_{i,1} = 1 \\ r_{i,2} = r_{i,3} = r_{i,4} = r_{i,5} = 0 \end{cases} \tag{4}$$

When $S_{i,j} \geq x_i \geq S_{i,j+1}$, the expression is:

$$\begin{cases} r_{i,j} = \frac{x_i - S_{i,j+1}}{S_{i,j} - S_{i,j+1}} \\ r_{i,j+1} = 1 - r_{i,j} \end{cases} \tag{5}$$

When $S_{i,5} \geq x_i$, the expression is:

$$\begin{cases} r_{i,5} = 1 \\ r_{i,1} = r_{i,2} = r_{i,3} = r_{i,4} = 0 \end{cases} \tag{6}$$

where $i$ is the evaluation factor number of water quality, $j$ is the water quality evaluation standard grade; $r_{i,j}$ is the membership degree of the $i-th$ factor to the $j-th$ water quality level; $S_{i,j}$ is the evaluation standard value of the $j-th$ factor of the $i-th$ factor; $x_i$ is the measured value of the $i-th$ factor.

According to above equations, the membership degree of each level standard with $i-th$ factors can be obtained. The fuzzy matrix $R$ is determined as below [31].

$$R = [r_{i,j}] = \begin{bmatrix} f_{11} & f_{12} & \cdots & f_{1n} \\ f_{21} & f_{22} & \cdots & f_{2n} \\ \cdots & \cdots & \cdots & \cdots \\ f_{m1} & \cdots & \cdots & f_{mn} \end{bmatrix} \tag{7}$$

Step 4: Construct the weight vector of evaluation factor [32]

The weight of evaluation factors reflects the status and role of each water quality factor in the process of water quality evaluation, and its calculation formula is as follows:

$$W = [w_1, w_2 \ldots w_i \ldots w_n] \tag{8}$$

where $w_i$ is the weight value of the $i-th$ evaluation factor.

Step 5: Comprehensive evaluation to make a decision

The fuzzy matrix $R$ is combined with the weight vector $W$ to obtain the comprehensive decision matrix $B$:

$$B = W \cdot R \tag{9}$$

*2.2. Improve Fuzzy Comprehensive Water Quality Assessment Method*

On the basis of the traditional fuzzy comprehensive water quality assessment method, the improved fuzzy comprehensive assessment method is mainly improved from the following two aspects:

(I) To construct an improved set of evaluation criteria according to the mainstream evaluation criteria currently in use;

(II) Consider that the water quality evaluation factor of the same aquaculture water area changes periodically with the seasons (transverse) [33]; In addition, in different years, in the same season and on the same day, the water quality assessment factors will be correlated (longitudinal), and the weight of water quality assessment factors will be reconstructed.

2.2.1. Construction of Improvement Evaluation Criteria Set
Water Quality Assessment Factor Set

The dissolved oxygen, pH value, temperature, ammonia nitrogen content, color and taste, and floating objects in the same aquaculture water were selected as evaluation factors to construct evaluation factor subset (evaluation factor subset is shown in Table 1).

$$U = \{\text{dissolved oxygen, pH value, temperature, ammonia nitrogen content, color and taste, float}\} \tag{10}$$

**Table 1.** Evaluation factor set.

| Serial Number | Evaluation Factor | Serial Number | Evaluation Factor |
|---|---|---|---|
| 1 | Dissolved oxygen | 2 | Temperature |
| 3 | pH | 4 | Ammonia nitrogen |
| 5 | Color taste | 6 | Floating object |

According to the traditional fuzzy comprehensive evaluation method, a certain water quality monitoring data of the same aquaculture water area is given weight:

$$W = [w_1, w_2 \ldots . w_i \ldots w_m] \tag{11}$$

where $W$ is the measured weight vector (that is, the measured factor's measured weight vector), and $w_i$ is the weight value of the $i - th$ evaluation factor.

Where:

$$w_i = \frac{x_i / \overline{S_i}}{\sum\limits_{i=1}^{3} \left( x_i / \overline{S_i} \right)}, i = 1, 2, 3 \tag{12}$$

$$\overline{S_i} = \frac{1}{k} \sum\limits_{i=1}^{k} S_{i,j}, i = 1, 2, 3 \tag{13}$$

where $k$ (In this paper, $k=4$) is the number of levels in the water quality evaluation standard set. $x_i$ is the measured value, $S_{i,j}$ is the standard value of the $i - th$ factor and $j - th$ level, and $\overline{S_i}$ is mean value of the full standard level of the $i - th$ factor.

Water Quality Evaluation Standards

This paper combines the Fishery Water Quality Standard with the Marine Water Quality Standard to construct a new evaluation standard for specific objects. Specific practices are as follows:

(I) Take the first two of the four categories in the Marine Water Quality Standard.

(II) Combining the first two types of evaluation standards of Seawater Quality Standard with fishery Water Quality Standard, the improved evaluation standards are divided into 4 categories. The set of improved evaluation criteria is shown in Table 2.

**Table 2.** Improved set of evaluation criteria.

| Evaluation Factor | Excellent (I) | Good (II) | | Qualified (III) | | Disqualified (IV) | |
|---|---|---|---|---|---|---|---|
| Dissolved Oxygen (mg/L) | [6, 8] | [5, 6) ∪ (8, 10] | | [3, 5) | | DO < 3 | |
| Represent-ative value (mg/L) | 7 | 5.5 | 9 | 4 | | 3 | |
| Temperature (°C) | [24, 28] | [22, 24) ∪ (28, 30] | | [20, 24) ∪ (30, 32] | | T < 20 or T > 32 | |
| Represent-ative value (°C) | 26 | 23 | | 29 | | 22 | |
| Ammonia Nitrogen (mg/L) | [0, 0.10] | (0.10, 0.15] | | (0.15, 0.20] | | AN > 0.20 | |
| Represent-ative value (mg/L) | 0.05 | 0.125 | | 0.175 | | 0.20 | |
| pH | [7, 8) | [6.5, 7) ∪ (8, 8.5] | | [6, 6.5) ∪ (8.5, 9) | | pH < 6 or pH > 9.0 | |
| Represent-ative value | 7.5 | 6.75 | 8.25 | 6.25 | 8.75 | 6 | 9 |
| Color, smelly, taste | Don't make fish, shrimp, shellfish, algae with abnormal colors, odors | | | | | | |
| floating object | No obvious oil film, floating foam and other floating substances on the water surface | | | | | | |

2.2.2. Time Dimension Water Quality Factor Weight Reconstruction

Improved Membership Function

The membership function commonly used in the traditional fuzzy comprehensive water quality evaluation method is incremental and decremental, these two membership functions have certain requirements for the size of the numerical value, and any one membership function cannot be applied to the whole system [18]. Therefore, this paper combines the above two membership functions to form an improved membership function, which is not limited by the numerical size and can be applied to the whole system. The improved membership function is shown below:

$$
f_{i1}(x_i) = \begin{cases} 1 & x_i \leq S_{i,1} \\ \frac{S_{i,2}-x_i}{S_{i,2}-S_{i,1}} & S_{i,1} < x_i < S_{i,2} \\ 0 & x_i \geq S_{i,2} \end{cases} \tag{14}
$$

$$
f_{i2}(x_i) = \begin{cases} 0 & x_i \leq S_{i,1}, x_i \geq S_{i,2} \\ \frac{x_i-S_{i,1}}{S_{i,2}-S_{i,1}} & S_{i,1} < x_i < S_{i,2} \\ \frac{S_{i,3}-x_i}{S_{i,3}-S_{i,2}} & S_{i,2} < x_i < S_{i,3} \\ 1 & x_i = S_{i,2} \end{cases} \tag{15}
$$

$$
f_{i3}(x_i) = \begin{cases} 0 & x_i \leq S_{i,2}, x_i \geq S_{i,4} \\ \frac{x_i-S_{i,2}}{S_{i,3}-S_{i,2}} & S_{i,2} < x_i < S_{i,3} \\ \frac{S_{i,4}-x_i}{S_{i,4}-S_{i,3}} & S_{i,3} < x_i < S_{i,4} \\ 1 & x_i = S_{i,3} \end{cases} \tag{16}
$$

$$
f_{i4}(x_i) = \begin{cases} 0 & x_i < S_{i,3} \\ \frac{x_i-S_{i,3}}{S_{i,4}-S_{i,3}} & S_{i,3} < x_i < S_{i,4} \\ 1 & x_i \geq S_{i,4} \end{cases} \tag{17}
$$

where $i$ is the evaluation factor number off water quality, $j$ is the water quality evaluation standard grade; $f_{i,j}$ is the membership degree of the $i$ factor corresponding to the $j$ level, $x_i$ is the measured value of each factor, and $S_{i,j}$ is the $i$ factor corresponding to the $j$ water quality level.

According to above-mentioned equations, the membership degree of the factor $i$ corresponding to each level of water quality level is obtained. Fuzzy matrix $R$ is constructed.

$$R = \begin{bmatrix} f_{11} & f_{12} & \cdots & f_{1j} \\ f_{21} & f_{22} & \cdots & f_{2j} \\ \cdots & \cdots & \cdots & \cdots \\ f_{i1} & f_{i2} & \cdots & f_{ij} \end{bmatrix} \tag{18}$$

Water Quality Evaluation Factor Weights

Taking the weight calculation method of a single monitoring point as an example, the evaluation factor weight is updated in two steps, that is, the horizontal weight vector and the vertical weight vector are updated.

(A). Update of Horizontal Weight Vector

In order to reflect the weight level of water quality assessment factors in different seasons, monitoring data of water quality assessment factors were divided into 4 groups according to the four seasons of spring, summer, autumn, and winter, forming 4 n × m matrices. Here, astronomical seasons and climatic seasons are combined to divide the four seasons, namely, March, April, and May are spring, June, July, and August are summer, September, October, and November are autumn, and December, January, and February are winter. The monitoring data matrix of one set (monitoring data of a certain season) is processed as follows:

$$U = \begin{bmatrix} u_{11} & u_{12} & \cdots & u_{1m} \\ u_{21} & u_{22} & \cdots & u_{2m} \\ \cdots & \cdots & \cdots & \cdots \\ u_{n1} & u_{n2} & \cdots & u_{nm} \end{bmatrix} \tag{19}$$

where $u_{i,j}$ represents the detection value of the $i - th$ detection index of the $j - th$ term.

The weighted average of the data of each evaluation factor is calculated, and the weighted average line vector of the variation of the monitoring value is obtained:

$$V = [v_j] = \begin{bmatrix} v_1 & v_2 & \cdots & v_m \end{bmatrix} \tag{20}$$

where,

$$v_j = \frac{\sum\limits_{i=1}^{n} u_{ij} * k_j}{\sum\limits_{i=1}^{n} k_j} \tag{21}$$

where $k_i$ is the number of occurrences of the water quality level to which the $i - th$ detection data belong.

For example, the temperature data in a test water quality is:

$$u_i = \begin{bmatrix} a \\ a \\ b \\ b \\ c \end{bmatrix} \tag{22}$$

where, $a$, $b$, and $c$ respectively indicate that the data of this test. According to the evaluation standard set, they are divided into $a$, $b$, and $c$ water quality levels, then:

$$k_a = 2, k_b = 2, k_c = 1 \tag{23}$$

$$\sum_{i=1}^{3} k_j = 2 + 2 + 1 = 5 \tag{24}$$

where, the number of times that $a$ single factor belongs to a water quality is 2, the number of times that it belongs to $b$ water quality is 2, and the number of times that it belongs to $c$ water quality is 1.

Calculate the weighted average relative change of the detected values, and construct a row vector of detected value changes:

$$V' = [v'_j] = [\ v'_1 \quad v'_2 \quad \dots \quad v'_m \ ] \tag{25}$$

where,

$$v'_j = \left| \sum_{i=1}^{m} \frac{v_i - v_j}{v_j} \right| \tag{26}$$

Let the arithmetic mean of a certain factor data in a certain test data be $M_1$ and the weighted mean be $M_2$. Scale factor:

$$m = \frac{M_1}{M_2} \tag{27}$$

Equation (26) can be simplified as:

$$v'_j = \left| \sum_{i=1}^{n} \frac{v_i - v_j}{v_j} \right| = \left| \frac{\sum_{i=1}^{n} v_i}{v_j} - n \right| = \left| \frac{n * M_1}{M_2} - n \right| = |n(m-1)| \tag{28}$$

Equation (28) can well reflect the weighted average variation of data. If the smaller data weight in a set of data is significant, then the single factor of the evaluation factor belongs to the water quality grade more times, and the weighted average will be smaller. If the larger data weight is significant, then the weighted average is larger. Therefore, the weighted average is changing.

When $v_j = 0$, it means there is no fluctuation in the detection data of this group. That is to say, the arithmetic $M_1$ and the weighted average $M_2$ representing the data of the group are equal. There are the following two cases:

(I) The number of occurrences of $k_j$ for each membership level is one.

(II) The frequency of the occurrence of $k_j$ for each membership grade is equal, which is a.

For the first case, because the selection of the evaluation standard set is divided into four levels, the availability of the algorithm can be guaranteed as long as the measurement data of the set is checked for at least 4 times.

For the second case, the conditions need to be satisfied:

$$\begin{cases} k_{j1} = k_{j2} = k_{j3} = k_{j4} = a \\ Condition2 \end{cases} \tag{29}$$

Condition2: Without loss of generality, set a certain set of data as:

$$u_{i,j} = \left[ u_{1,1} \dots u_{a,1}, u_{(a+1),2} \dots u_{2a,2}, u_{(2a+1),3} \dots u_{3a,3}, u_{(3a+1),4} \dots u_{4a,4} \right]^T \tag{30}$$

where $u_{i,j}$ represents the value of the $i-th$ detection. The single factor belongs to the $j-th$ water quality. Obviously, there is:

$$k_{j1} = k_{j2} = k_{j3} = k_{j4} = a \tag{31}$$

The arithmetic average and weighted average of this group of data are respectively:
Arithmetic average:

$$M_1 = \frac{\sum_{i=1}^{4a} u_i}{4a} \tag{32}$$

Weighted average:

$$M_2 = \frac{\sum\limits_{i=1}^{n} u_i * k_j}{\sum\limits_{i=1}^{n} k_j} = \frac{\sum\limits_{i=1}^{a} u_i * a + \sum\limits_{i=a+1}^{2a} u_i * a + \sum\limits_{i=2a+1}^{3a} u_i * a + \sum\limits_{i=3a+1}^{4a} u_i * a}{4a^2} = \frac{\sum\limits_{i=1}^{4a} u_i}{4a} \quad (33)$$

It can be obtained that when the number of measurement data is more than 4 times, if the data fluctuates, the weighted change may be 0. For this situation, the following requirements should be met:

$$k > 4 \text{ and } k \neq 4a \quad (34)$$

where $k$ is the sample size of the single factor data, and a is an integer. Actually, time is mainly divided according to months. Except for the non-leap year, which has 28 days in February, the days in the remaining months are not multiples of 4, and even in February of a non-leap year, it is a small probability event that a single factor belongs to the same number of water quality levels. It does not affect the practicality of the algorithm.

Construct the judgment matrix:

$$D = \left[ d_{ij} \right] = \begin{bmatrix} d_{11} & d_{12} & \dots & d_{1m} \\ d_{21} & d_{22} & \dots & d_{2m} \\ \dots & \dots & \dots & \dots \\ d_{m1} & d_{m2} & \dots & d_{mm} \end{bmatrix} \quad (35)$$

where,

$$d_{ij} = \frac{v_i'}{v_j'} \quad (36)$$

Obtain the feature vector:

$$A_0' = [a_{0i}'] \quad (37)$$

Obtain the weight vector after processing:

$$W' = \begin{bmatrix} w_1' & w_2' & \dots & w_m' \end{bmatrix} \quad (38)$$

where,

$$w_i' = \frac{a_{0i}'}{\sum\limits_{i=1}^{m} a_{0i}'} \quad (39)$$

The obtained feature vector is the lateral weight vector we need. At this point, the lateral weight vector is updated.

(B).　Update of Vertical Weight Vector

Considering that the longitudinal weight vector represents the deep mining of data. In the selection of data, according to a certain period, that is, each natural period in the same order of time. The calculation is based on the monitoring data of the same day of each year.

Construct the measured data matrix:

$$U'' = \left[ u_{ij}' \right] = \begin{bmatrix} u_{11}' & \dots & u_{1m}' \\ u_{21}' & \dots & u_{2m}' \\ \dots & \dots & \dots \\ u_{n1}' & \dots & u_{nm}' \end{bmatrix} \quad (40)$$

where, $u_{i,j}$ is the value of the $j-th$ evaluation factor on the same day in the $i-th$ year.

Based on the measured data, a variance matrix is constructed:

$$\sigma^2 = \left[\sigma_{ij}^2\right] = \begin{bmatrix} \sigma_{11}^2 & \sigma_{12}^2 & \cdots & \sigma_{1m}^2 \\ \sigma_{21}^2 & \cdots & \cdots & \sigma_{2m}^2 \\ \cdots & \cdots & \cdots & \cdots \\ \sigma_{n1}^2 & \sigma_{n2}^2 & \cdots & \sigma_{nm}^2 \end{bmatrix} \tag{41}$$

where $\sigma_{ij}^2$ represents the overall variance of the $i-th$ year before the $j-th$ evaluation factor.

$$\sigma_{ij}^2 = \frac{\sum\limits_{l=1}^{i} (u_l - E_i)^2}{i} \tag{42}$$

where, $E_i$ represents the mean value of $i$ years before the data.

$$E_i = \frac{\sum\limits_{l=1}^{i} u_l}{i} \tag{43}$$

The variance matrix reflects the data fluctuation of the same place on the same day in different years. The closer $\sigma_{ij}^2$ is to 0, the more stable the historical climate at the point is.

The principle is the same as the steps when constructing the horizontal weight vector. The judgment matrix is constructed as follows:

$$D' = [d_{kl}'] = \begin{bmatrix} d_{11}' & d_{12}' & \cdots & d_{1m}' \\ d_{21}' & d_{22}' & \cdots & d_{2m}' \\ \cdots & \cdots & \cdots & \cdots \\ d_{m1}' & d_{m2}' & \cdots & d_{mm}' \end{bmatrix} \tag{44}$$

where,

$$d_{kl}' = \frac{\sigma_k^2}{\sigma_l^2} \tag{45}$$

where, $\sigma_k^2$ is the average population variance of the $k-th$ factor in the first n years, that is:

$$\sigma_k^2 = \frac{\sum\limits_{i=1}^{n} \sigma_i^2}{n} \tag{46}$$

Considering the significance of practical application, the judgment matrix $D'$ is modified as follows:

$$\overline{D}' = [d_{kl}'] = \begin{bmatrix} 1 & 0 & 0 & \cdots & 0 \\ 1 & 1 & d_{23}' & \cdots & d_{2m}' \\ 1 & d_{32}' & \cdots & \cdots & d_{3m}' \\ \cdots & \cdots & \cdots & 1 & \cdots \\ 1 & d_{n2}' & d_{n3}' & \cdots & 1 \end{bmatrix} \tag{47}$$

The remaining steps are similar to solving the lateral weight vector. The feature vector is:

$$W_0'' = [w_{0i}''] \tag{48}$$

After processing, the weight vector of each evaluation factor is obtained:

$$W'' = \begin{bmatrix} w_1'' & w_2'' & \cdots & w_m'' \end{bmatrix} \tag{49}$$

where,

$$w_i'' = \frac{w_{0i}''}{\sum\limits_{i=1}^{m} w_{0i}''} \tag{50}$$

The resulting weight vector is the longitudinal weight vector we need. At this point, the vertical weight vector is updated.

Three evaluation factor weight vectors are combined to obtain the comprehensive weight vector of evaluation factor.

The measured weight vector is:

$$W = \begin{bmatrix} w_1 & w_2 & \ldots & w_m \end{bmatrix} \tag{51}$$

The horizontal weight vector is:

$$W' = \begin{bmatrix} w_1' & w_2' & \ldots & w_m' \end{bmatrix} \tag{52}$$

The vertical weight vector is:

$$W'' = \begin{bmatrix} w_1'' & w_2'' & \ldots & w_m'' \end{bmatrix} \tag{53}$$

The comprehensive weight vector is calculated is:

$$W''' = \begin{bmatrix} w_1''' & w_2''' & \ldots & w_m''' \end{bmatrix} \tag{54}$$

where,

$$w_i''' = \frac{w_i \cdot w_i' \cdot w_i''}{\sum\limits_{k=1}^{m} \left( w_k \cdot w_k' \cdot w_k'' \right)} \tag{55}$$

Fuzzy comprehensive evaluation *B* is performed:

$$B = W \cdot R = W''' \cdot R \tag{56}$$

## 3. Application Examples

The location we choose is the aquaculture base in Yangjiapo town, Binhai New Area, Tianjin, a coastal city in northern China, as shown in Figure 1. Yangjiabo town has a typical temperate monsoon climate, which is characterized by windy, sunny and suitable temperature in spring, summer and autumn; cold and foggy in winter. Therefore, in the spring, summer, and autumn, the water flow velocity in the aquaculture area is relatively high and the water surface temperature is mild, while in winter, the water surface velocity is slow. The main breeding species in an aquaculture base are nuisanceless white carp, grass carp, and crucian carp. Aquaculture waters have sufficient water sources and relevant water quality assurance measures are in place to ensure water quality. Aquaculture pond area 0.67 hectares, pool water depth 2.5 m, rectangular fishing pond, length to width ratio of 5:3, east–west trend, pond ridge leakage. In order to ensure the quality of aquaculture water, it is necessary to monitor and evaluate the water quality of aquaculture ponds. For this reason, the water quality of the pond has been monitored for three years since March 2017. According to the water quality requirements of pond culture, dissolved oxygen, pH value, temperature, and ammonia nitrogen content were selected as the main monitoring indexes. Water sample testing was completed on the sampling day, and the national standard method was adopted for the determination of the three indexes. Table 3 shows some results of water quality monitoring (Serial number 1–20 is the measured values of spring, summer, autumn, and winter from March 2016 to February 2017; Serial number 21–40 is the measured value of spring, summer, autumn, and winter from March 2017 to February 2018. Serial number 41–60 is the measured values of spring, summer, autumn, and winter from March 2018 to February 2019. Take 5 typical data in each season).

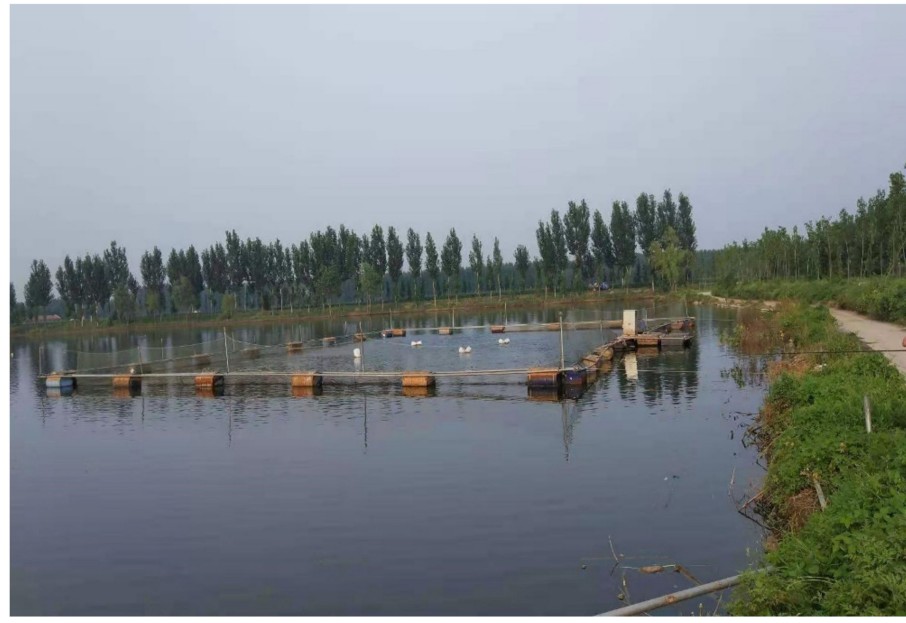

**Figure 1.** Yangjiapo Aquaculture Base, Binhai New Area, Tianjin, China.

**Table 3.** Test data sheet.

| Serial Number | Temperature | Dissolved Oxygen | pH | Serial Number | Temperature | Dissolved Oxygen | pH |
|---|---|---|---|---|---|---|---|
| 1 | 20.3 | 4.7 | 6.8 | 31 | 20.4 | 7.3 | 6.2 |
| 2 | 20.4 | 4.7 | 6.7 | 32 | 20.5 | 7.3 | 6.3 |
| 3 | 20.4 | 4.6 | 6.5 | 33 | 20.4 | 7.3 | 6.3 |
| 4 | 20.3 | 4.7 | 6.6 | 34 | 20.2 | 7.2 | 6.2 |
| 5 | 20.2 | 4.4 | 6.7 | 35 | 20.3 | 7.2 | 6.4 |
| 6 | 20.6 | 4.5 | 6.6 | 36 | 19.9 | 7.1 | 6.1 |
| 7 | 20.2 | 4.7 | 6.6 | 37 | 20.1 | 6.9 | 6.2 |
| 8 | 20.1 | 4.8 | 6.7 | 38 | 20.0 | 6.6 | 6.1 |
| 9 | 20.3 | 4.8 | 6.4 | 39 | 20.2 | 6.8 | 6.0 |
| 10 | 20.5 | 4.6 | 6.7 | 40 | 20.1 | 6.8 | 6.1 |
| 11 | 28.2 | 4.4 | 7.1 | 41 | 20.0 | 6.6 | 6.2 |
| 12 | 26.9 | 4.3 | 7.3 | 42 | 19.9 | 6.6 | 6.3 |
| 13 | 25.7 | 4.4 | 7.2 | 43 | 20.1 | 6.7 | 6.3 |
| 14 | 24.6 | 4.5 | 7.1 | 44 | 19.9 | 6.4 | 6.2 |
| 15 | 23.6 | 4.3 | 7.3 | 45 | 19.8 | 6.5 | 6.1 |
| 16 | 22.7 | 4.6 | 6.9 | 46 | 19.9 | 6.3 | 6.1 |
| 17 | 21.9 | 4.7 | 6.6 | 47 | 19.9 | 6.5 | 6.0 |
| 18 | 21.2 | 4.6 | 6.4 | 48 | 20.0 | 6.4 | 6.2 |
| 19 | 20.6 | 4.5 | 6.8 | 49 | 20.1 | 6.3 | 6.1 |
| 20 | 20.5 | 4.6 | 6.6 | 50 | 20.1 | 6.4 | 6.2 |
| 21 | 20.4 | 4.4 | 6.5 | 51 | 20.5 | 6.2 | 7.6 |
| 22 | 19.8 | 4.5 | 6.5 | 52 | 20.6 | 6.2 | 7.6 |
| 23 | 20.2 | 4.7 | 6.4 | 53 | 20.6 | 6.1 | 7.5 |
| 24 | 20.3 | 4.5 | 6.4 | 54 | 20.5 | 6.2 | 7.6 |
| 25 | 20.1 | 4.6 | 6.3 | 55 | 20.4 | 6.1 | 7.4 |
| 26 | 20.2 | 4.3 | 6.2 | 56 | 20.3 | 6.0 | 7.5 |
| 27 | 20.2 | 4.2 | 6.2 | 57 | 20.3 | 6.0 | 7.5 |
| 28 | 20.0 | 4.3 | 6.1 | 58 | 20.2 | 5.9 | 7.4 |
| 29 | 20.2 | 4.4 | 6.3 | 59 | 20.2 | 5.8 | 7.4 |
| 30 | 19.9 | 4.6 | 6.1 | 60 | 19.9 | 5.8 | 7.4 |

The evaluation criteria for water quality are divided into four grades, namely I, II, III, and IV. Among them, the water quality of grade I is the best, and gradually deteriorates as the grade increases, while grade IV is the worst. The evaluation results of the single factor evaluation method, the traditional fuzzy comprehensive evaluation method, and the improved traditional fuzzy evaluation method are shown in Tables 4–6, respectively (the results remain three decimal places). By comparing the results table, it can be seen that that when using the single factor evaluation method, the annual water quality grade of the aquaculture pond corresponds to the worst grade of the factor, most of which are grade III, that is, the water quality is poor (Table 4); when using the traditional evaluation method, the water quality grade tends to correspond with the level of the evaluation factor

with mutation content changes, and only in the autumn is a higher grade, which can reach grade I or grade II, while the grade is generally low in other seasons (Table 5); when using with the improved method, aquaculture ponds in winter state of water quality is poorer, mostly for level III water quality grade, in spring, summer, autumn state of water quality is better, water quality grade for class I generally, in spring and summer and autumn part of the water quality grade for class II, prove the breeding pond water quality state in spring and summer have more obvious improvement(Table 6).

**Table 4.** Single factor evaluation method water quality scale.

| Serial Number | Temperature | Dissolved Oxygen | pH | Water Quality Grade | Serial Number | Temperature | Dissolved Oxygen | pH | Water Quality Grade |
|---|---|---|---|---|---|---|---|---|---|
| 1 | III | III | II | III | 31 | III | I | III | III |
| 2 | III | III | II | III | 32 | III | I | III | III |
| 3 | III | III | II | III | 33 | III | I | III | III |
| 4 | III | III | II | III | 34 | III | I | III | III |
| 5 | III | III | II | III | 35 | III | I | III | III |
| 6 | III | III | II | III | 36 | IV | I | III | IV |
| 7 | III | III | II | III | 37 | III | I | III | III |
| 8 | III | III | II | III | 38 | IV | I | III | IV |
| 9 | III | III | III | III | 39 | III | I | III | III |
| 10 | III | III | II | III | 40 | III | I | III | III |
| 11 | II | III | I | III | 41 | IV | I | III | IV |
| 12 | I | III | I | III | 42 | IV | I | III | IV |
| 13 | I | III | I | III | 43 | III | I | III | III |
| 14 | I | III | I | III | 44 | IV | I | III | IV |
| 15 | III | III | I | III | 45 | IV | I | III | IV |
| 16 | III | III | II | III | 46 | IV | I | III | IV |
| 17 | III | III | II | III | 47 | IV | I | III | IV |
| 18 | III | III | III | III | 48 | IV | I | III | IV |
| 19 | III | III | II | III | 49 | III | I | III | III |
| 20 | III | III | II | III | 50 | III | I | III | III |
| 21 | III | III | II | III | 51 | III | I | I | III |
| 22 | IV | III | II | IV | 52 | III | I | I | III |
| 23 | III | III | III | III | 53 | III | I | I | III |
| 24 | III | III | III | III | 54 | III | I | I | III |
| 25 | III | III | III | III | 55 | III | I | I | III |
| 26 | III | III | III | III | 56 | III | I | I | III |
| 27 | III | III | III | III | 57 | III | I | I | III |
| 28 | IV | III | III | IV | 58 | III | II | I | III |
| 29 | III | III | III | III | 59 | III | II | I | III |
| 30 | IV | III | III | IV | 60 | IV | II | I | IV |

**Table 5.** Tradition fuzzy comprehensive water quality evaluation membership degree table.

| N | r1 | r2 | r3 | r4 | L | N | r1 | r2 | r3 | r4 | L |
|---|---|---|---|---|---|---|---|---|---|---|---|---|
| 1 | 0.024 | 0.490 | 0.229 | 0.257 | II | 31 | 0.392 | 0.069 | 0.274 | 0.265 | I |
| 2 | 0.000 | 0.476 | 0.280 | 0.244 | II | 32 | 0.390 | 0.097 | 0.318 | 0.196 | I |
| 3 | 0.000 | 0.310 | 0.441 | 0.249 | III | 33 | 0.390 | 0.097 | 0.305 | 0.209 | I |
| 4 | 0.000 | 0.405 | 0.335 | 0.260 | II | 34 | 0.413 | 0.046 | 0.250 | 0.291 | I |
| 5 | 0.000 | 0.411 | 0.310 | 0.279 | II | 35 | 0.409 | 0.131 | 0.239 | 0.221 | I |
| 6 | 0.000 | 0.357 | 0.424 | 0.219 | III | 36 | 0.436 | 0.023 | 0.112 | 0.429 | I |
| 7 | 0.000 | 0.406 | 0.320 | 0.274 | II | 37 | 0.419 | 0.030 | 0.242 | 0.310 | I |
| 8 | 0.000 | 0.501 | 0.214 | 0.285 | II | 38 | 0.323 | 0.118 | 0.115 | 0.444 | IV |
| 9 | 0.000 | 0.291 | 0.448 | 0.261 | III | 39 | 0.389 | 0.060 | 0.027 | 0.524 | IV |
| 10 | 0.000 | 0.452 | 0.317 | 0.231 | II | 40 | 0.388 | 0.060 | 0.127 | 0.426 | IV |
| 11 | 0.256 | 0.531 | 0.213 | 0.000 | II | 41 | 0.321 | 0.117 | 0.233 | 0.328 | IV |
| 12 | 0.510 | 0.260 | 0.230 | 0.000 | I | 42 | 0.320 | 0.146 | 0.266 | 0.268 | I |
| 13 | 0.527 | 0.254 | 0.219 | 0.000 | I | 43 | 0.352 | 0.117 | 0.277 | 0.255 | I |
| 14 | 0.345 | 0.448 | 0.207 | 0.000 | II | 44 | 0.259 | 0.173 | 0.237 | 0.332 | IV |
| 15 | 0.333 | 0.426 | 0.241 | 0.000 | II | 45 | 0.292 | 0.146 | 0.116 | 0.446 | IV |
| 16 | 0.069 | 0.637 | 0.294 | 0.000 | II | 46 | 0.229 | 0.201 | 0.118 | 0.453 | IV |
| 17 | 0.000 | 0.395 | 0.589 | 0.016 | III | 47 | 0.293 | 0.146 | 0.000 | 0.561 | IV |
| 18 | 0.000 | 0.238 | 0.634 | 0.128 | III | 48 | 0.258 | 0.172 | 0.237 | 0.333 | IV |
| 19 | 0.024 | 0.444 | 0.315 | 0.217 | II | 49 | 0.228 | 0.200 | 0.131 | 0.440 | IV |
| 20 | 0.000 | 0.380 | 0.387 | 0.233 | III | 50 | 0.258 | 0.172 | 0.250 | 0.320 | IV |
| 21 | 0.000 | 0.264 | 0.483 | 0.252 | III | 51 | 0.480 | 0.255 | 0.066 | 0.199 | I |
| 22 | 0.000 | 0.289 | 0.404 | 0.307 | III | 52 | 0.480 | 0.255 | 0.080 | 0.186 | I |
| 23 | 0.000 | 0.267 | 0.456 | 0.277 | III | 53 | 0.497 | 0.235 | 0.081 | 0.188 | I |
| 24 | 0.000 | 0.217 | 0.516 | 0.267 | III | 54 | 0.480 | 0.255 | 0.066 | 0.199 | I |
| 25 | 0.000 | 0.174 | 0.531 | 0.295 | III | 55 | 0.451 | 0.281 | 0.054 | 0.214 | I |
| 26 | 0.000 | 0.067 | 0.575 | 0.358 | III | 56 | 0.474 | 0.259 | 0.040 | 0.227 | I |
| 27 | 0.000 | 0.044 | 0.596 | 0.360 | III | 57 | 0.474 | 0.259 | 0.040 | 0.227 | I |
| 28 | 0.000 | 0.068 | 0.407 | 0.525 | III | 58 | 0.401 | 0.329 | 0.027 | 0.242 | I |
| 29 | 0.000 | 0.125 | 0.590 | 0.285 | III | 59 | 0.377 | 0.352 | 0.027 | 0.244 | I |
| 30 | 0.000 | 0.142 | 0.346 | 0.512 | IV | 60 | 0.378 | 0.354 | 0.000 | 0.268 | I |

**Table 6.** Improved fuzzy comprehensive water quality evaluation membership degree table.

| N | r1 | r2 | r3 | r4 | L | N | r1 | r2 | r3 | r4 | L |
|---|---|---|---|---|---|---|---|---|---|---|---|
| 1 | 0.025 | 0.495 | 0.229 | 0.259 | I | 31 | 0.392 | 0.069 | 0.276 | 0.266 | I |
| 2 | 0.001 | 0.482 | 0.283 | 0.254 | I | 32 | 0.390 | 0.094 | 0.328 | 0.196 | I |
| 3 | 0.000 | 0.315 | 0.441 | 0.269 | I | 33 | 0.391 | 0.098 | 0.308 | 0.212 | II |
| 4 | 0.000 | 0.410 | 0.335 | 0.262 | I | 34 | 0.415 | 0.046 | 0.252 | 0.291 | I |
| 5 | 0.002 | 0.412 | 0.310 | 0.279 | I | 35 | 0.405 | 0.136 | 0.239 | 0.230 | I |
| 6 | 0.000 | 0.359 | 0.424 | 0.229 | I | 36 | 0.436 | 0.023 | 0.130 | 0.429 | III |
| 7 | 0.000 | 0.409 | 0.320 | 0.274 | I | 37 | 0.419 | 0.030 | 0.242 | 0.310 | II |
| 8 | 0.001 | 0.504 | 0.217 | 0.275 | I | 38 | 0.331 | 0.118 | 0.115 | 0.451 | II |
| 9 | 0.000 | 0.295 | 0.448 | 0.261 | I | 39 | 0.389 | 0.064 | 0.027 | 0.524 | III |
| 10 | 0.000 | 0.455 | 0.317 | 0.234 | I | 40 | 0.387 | 0.059 | 0.124 | 0.426 | III |
| 11 | 0.234 | 0.534 | 0.215 | 0.000 | I | 41 | 0.321 | 0.117 | 0.235 | 0.328 | I |
| 12 | 0.512 | 0.263 | 0.230 | 0.000 | I | 42 | 0.320 | 0.144 | 0.266 | 0.268 | I |
| 13 | 0.427 | 0.214 | 0.219 | 0.000 | II | 43 | 0.332 | 0.117 | 0.287 | 0.255 | I |
| 14 | 0.345 | 0.450 | 0.217 | 0.000 | I | 44 | 0.259 | 0.175 | 0.237 | 0.331 | I |
| 15 | 0.334 | 0.423 | 0.241 | 0.001 | I | 45 | 0.289 | 0.146 | 0.114 | 0.446 | I |
| 16 | 0.072 | 0.632 | 0.286 | 0.000 | III | 46 | 0.229 | 0.200 | 0.118 | 0.443 | I |
| 17 | 0.000 | 0.393 | 0.589 | 0.013 | II | 47 | 0.293 | 0.146 | 0.000 | 0.561 | I |
| 18 | 0.010 | 0.234 | 0.642 | 0.125 | II | 48 | 0.260 | 0.172 | 0.237 | 0.333 | I |
| 19 | 0.024 | 0.447 | 0.315 | 0.217 | III | 49 | 0.228 | 0.200 | 0.135 | 0.440 | I |
| 20 | 0.000 | 0.383 | 0.376 | 0.233 | III | 50 | 0.258 | 0.172 | 0.250 | 0.320 | I |
| 21 | 0.000 | 0.252 | 0.483 | 0.252 | I | 51 | 0.482 | 0.255 | 0.068 | 0.199 | I |
| 22 | 0.020 | 0.269 | 0.414 | 0.306 | I | 52 | 0.480 | 0.255 | 0.080 | 0.186 | I |
| 23 | 0.010 | 0.256 | 0.456 | 0.277 | I | 53 | 0.497 | 0.235 | 0.081 | 0.188 | II |
| 24 | 0.000 | 0.224 | 0.526 | 0.266 | I | 54 | 0.480 | 0.254 | 0.066 | 0.199 | I |
| 25 | 0.020 | 0.178 | 0.531 | 0.295 | I | 55 | 0.451 | 0.281 | 0.057 | 0.215 | I |
| 26 | 0.010 | 0.073 | 0.575 | 0.355 | I | 56 | 0.474 | 0.259 | 0.040 | 0.227 | III |
| 27 | 0.000 | 0.044 | 0.596 | 0.360 | I | 57 | 0.474 | 0.259 | 0.040 | 0.227 | II |
| 28 | 0.000 | 0.068 | 0.407 | 0.525 | I | 58 | 0.401 | 0.329 | 0.027 | 0.245 | II |
| 29 | 0.000 | 0.126 | 0.592 | 0.284 | I | 59 | 0.377 | 0.352 | 0.027 | 0.245 | III |
| 30 | 0.000 | 0.143 | 0.346 | 0.512 | I | 60 | 0.378 | 0.361 | 0.000 | 0.271 | III |

## 4. Discussion

In the single factor water quality evaluation method, the overall water quality level only corresponds to the worst evaluation level in the set of evaluation factors (Table 4), and cannot reflect the impact of other evaluation factors on water quality [34], which leads to deviations between the experimental results and the actual situation. The traditional fuzzy evaluation method selects fuzzy matrix and evaluation factor weights to construct a water quality evaluation model [35]. When the content of each water quality evaluation factor does not change drastically, the water quality grade depends on the weight of each water quality evaluation factor. At this time, the evaluation result is not only related to the selected evaluation standard, but also related to the calculation method of the weight. The overall idea is that when the content of one or more water quality evaluation factors changes dramatically, it is considered that the weight of the evaluation factor is higher than other evaluation factors at this moment, so the result will be more toward the evaluation factor of this content mutation (Table 5). The improved fuzzy water quality evaluation method proposed in this paper considers the relationship between temporal and spatial changes and water quality, and reflects the degree of water quality changing with the seasons. Xu Shiguo et al., established the membership function through the Nor-Half Sinusoidal Distribution Method, and used the improved fuzzy evaluation method mentioned above to described the water quality of the Nansi Lke in China in each season from the perspective of temporal and spatial distribution [36]. Meanwhile, in order to evaluation water quality more accurately, Liu Yu et al., evaluated the water quality of Yongding New River in Tianjin, China based on the comprehensive evaluation of improved water quality. The results are in line with reality and provide scientific support for water environmental management. Considering the impact of time changes on water quality is conducive to better governance

of water resources [37]. Ma Zhen et al., considered the concept of time and space and used the improved water quality index to evaluate the water quality of the aquaculture area on the south bank of Dalian, China. The results showed that the improved method had effective practicability and can be used for the time-space analysis of water quality in the Dalian aquaculture area [38].

In order to clearly express the impact of the improved method on the evaluation results, according to the comparison with the single factor water quality evaluation method and traditional fuzzy water quality evaluation method, this paper draws a contrast curve chart, as shown in Figure 2. It can be seen from Figure 2 that in the 60 measurements of the single factor water quality evaluation method, most of the water quality evaluation grade are level III or IV. The test results of the traditional fuzzy comprehensive evaluation method fluctuate greatly. While after improving the method, in the 30th to 50th test data, the content of each evaluation factor does not have a big mutation, but within a reasonable range, small fluctuations cause the evaluation result to have multiple mutations in this interval.

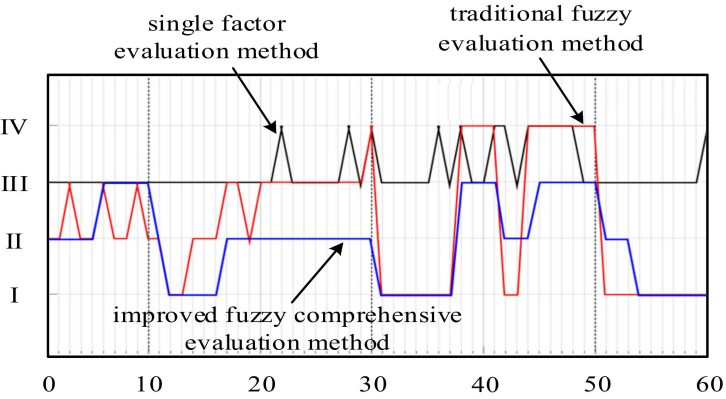

**Figure 2.** Comparison chart of evaluation results.

Through the comparison of the evaluation results, it can be seen that compared with the single factor water quality evaluation method and the traditional fuzzy comprehensive evaluation method, the improved fuzzy comprehensive evaluation method proposed in this paper considers the influence of the weight of the water quality evaluation factor and the time dimension on the water quality evaluation; what's more, this method combine the *Sea Water Quality Standard* and *Fishery Water Quality Standard* and improves the set of evaluation standards; in addition, in the same season and same day in different years, the status of the water quality evaluation factors are different. Therefore, the improved method proposed in this paper takes the periodic changes of the water quality evaluation factors in the same aquaculture water into consideration in the time dimension, and the weight of the evaluation factor is reconstructed. By using the improved fuzzy evaluation method to evaluate the water quality of the aquaculture area, and the evaluation results better reflect the dynamic change characteristics of the water quality of the aquaculture water area, which is in line with reality. It can be seen from Table 6 that the water quality in the aquaculture area changes with the seasons. The water quality in winter is poor, and the water quality in the rest of the season has improved significantly. This has important reference value for future analysis of water quality changes and water quality management.

## 5. Conclusions

Aiming at the accuracy of aquaculture water quality evaluation, and taking into account the impact of different years and different seasons on the evaluation results, this paper has made improvements on the basis of the traditional fuzzy comprehensive evaluation method: firstly, we combined the *Sea water Quality Standard* and *Fishery Water Quality Standard* to improve the evaluation standard set; secondly, by combining the increasing and decreasing membership functions of the traditional fuzzy comprehensive

evaluation method, we constructed an improved membership function, and obtained the evaluation factor membership fuzzy matrix; finally, taking the weight calculation method of a single monitoring point as an example, we reconstructed the time domain weight matrix of the evaluation factor in two steps to obtain the comprehensive weight vector, and then perform the evaluation.

The improved fuzzy comprehensive evaluation method reflects the changes of the monitoring data of each index over time, and can reflect the difference in importance of each index in different years and seasons. The comprehensive weight vector can not only reflect the difference of various index over time, but also reflect the real-time weight of water quality index. This paper takes into consideration the cyclical changes of water quality factors with the four seasons and the correlation of water quality factors on the same day in different years, and use an improved method to comprehensively evaluate the water quality of the Yangjiabo Aquaculture Base in Tianjin, China. The results show that compared with the single factor evaluation method and the traditional fuzzy comprehensive water quality evaluation method, the improved method can more accurately reflect the dynamic change characteristics of the water quality in the aquaculture area, and the specific improvement of the water quality in spring, summer and autumn compared with winter. Consequently, it can be said that the results are more in line with the reality.

In conclusion, a more accurate and comprehensive water quality assessment is conducive to strengthen water management and improve the efficiency of aquaculture. The research method and results of this paper can provide certain reference in the research of aquaculture water quality evaluation, aquaculture and water management.

**Author Contributions:** Conceptualization, G.Y. and B.X.; methodology, G.Y.; software, H.S.; validation, G.Y., B.X. and S.Z.; formal analysis, J.P.; investigation, H.S.; resources, J.L.; data curation, X.H.; writing—original draft preparation, G.Y.; writing—review and editing, G.Y.; visualization, B.X.; supervision, R.D.; project administration, G.Y.; funding acquisition, G.Y. All authors have read and agreed to the published version of the manuscript.

**Funding:** This project was funded by the Tianjin Science and Technology Support Foundation of China (No.17YFZCNC00230) and the Key Laboratory of Equipment and Informatization in Environment Controlled Agriculture, Ministry of Agriculture and Rural Affairs, China (No. 2011NYZD1902).

**Institutional Review Board Statement:** Not applicable.

**Informed Consent Statement:** Not applicable.

**Data Availability Statement:** The data supporting the findings of this study are available from the corresponding author, upon reasonable request.

**Acknowledgments:** This research was made possible thanks to the Tianjin Science and Technology Support Foundation of China and the Key Laboratory of Equipment and Informatization in Environment Controlled Agriculture, Ministry of Agriculture and Rural Affairs, China.

**Conflicts of Interest:** All authors have no conflict of interest regarding this manuscript.

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
