# Peer review of "Evaluation of Aquaculture Water Quality Based on Improved Fuzzy Comprehensive Evaluation Method"

_water, doi:10.3390/w13081019_

Round 1

Reviewer 1 Report

This manuscript structure is not suitable for an engineering journal. It is better author submit it at a mathematics journal.

Also, novelty of this study is not clear.

Reviewer 2 Report

Authors have provided very interesting and useful application of the fuzzy logic method for evaluation of aquaculture water quality regarding particular quality parameters. Methodology is well developed and explained in detail, as well as defuzzification. I am proposing a major revision due to some lack of explanations.

1) Why have authors selected these particular water quality parameters for testing [4-6]? This is only for the case study, i.e. observed location in China?2) Why haven't the authors described the observed location? There is a lack of map, site description, pictures of the microlocation...

3) Subchapter ''4.1. Related studies on water quality evaluation'' does not have the right place in the Discussion. This should be incorporated into the review section/introduction.

4) English language and grammar is at a low level. Authors should handle this. 

Reviewer 3 Report

- very interesting manuscript

Round 2

Reviewer 1 Report

Abstract is not fit for this manuscript, please re-write it again.

Materials and method section needs more references.

Discussion part should be extended.

Tables 4 and 5 need more explanation.

Provide some recommendation in Conclusion section.

Reviewer 2 Report

I am suggesting a publishing of the paper. Authors have done changes in accordance to my review. Also, other reviews are satisfied, as I can see. 

Round 3

Reviewer 1 Report

This manuscript is acceptable.

This manuscript is a resubmission of an earlier submission. The following is a list of the peer review reports and author responses from that submission.